# The Effect of 5-Aminolevulinic Acid Photodynamic Therapy in Promoting Pyroptosis of HPV-Infected Cells

**Junxiao Wei, Xiaoming Peng, Sijia Wang, Meinian Xu, Hui Liu, Yixiu Zhong, Xi Chen, Qi Wang, Xiaowen Huang \* and Kang Zeng \***

Department of Dermatology and Venereology, Nanfang Hospital, Southern Medical University, Guangzhou 510515, China; wjxgdgz@163.com (J.W.); oliver1986@126.com (X.P.); wangsj517@163.com (S.W.); xumn1993@163.com (M.X.); nhdx2016liuhui@163.com (H.L.); zuyy5@i.smu.edu.cn (Y.Z.); xchencc@163.com (X.C.); wangqi1987@smu.edu.cn (Q.W.)
\* Correspondence: hxw617@163.com (X.H.); nfpfkzk@126.com (K.Z.)

**Abstract:** 5-aminolevulinic acid photodynamic therapy (ALA-PDT) is highly effective in the treatment of condyloma acuminata (CA). Previous research has indicated that ALA-PDT could induce cell death by different mechanisms, including apoptosis and autophagy, but the role of pyroptosis in ALA-PDT remains uncertain. Thus, this study aimed to explore whether pyroptosis is a potential mechanism of ALA-PDT killing human papillomavirus (HPV) infected cells. HPV-positive HeLa cells were exposed to ALA-PDT, then cell viability assay, lactate dehydrogenase release (LDH) assay, detection of reactive oxygen species (ROS), quantitative real-time PCR (qPCR), and western blot were used to evaluate pyroptosis induced by ALA-PDT. Results suggested that ALA-PDT enhanced the expression of NLRP3, caspase-1, GSDMD, and the production of inflammatory cytokines such as IL-1β and IL-18. In addition, ALA-PDT induced the production of ROS and led to the destruction of the cell membrane. The inhibition of pyroptosis reduced the killing of HeLa cells by ALA-PDT. This study demonstrates that ALA-PDT induces pyroptosis in HPV-positive cells, which provides some explanation for the mechanism of ALA-PDT to treat CA and HPV infection-related diseases.

**Keywords:** photodynamic therapy; pyroptosis; HPV infection; condyloma acuminata

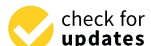



## 1. Introduction

Human papillomavirus (HPV) is a highly contagious virus that causes common warts, condyloma acuminata (CA), and increases the risk of cervical or genitourinary cancer. CA is mostly caused by low-risk (LR) HPV6 and 11, showing a trend in increase worldwide and becoming one of the most common sexually transmitted infections in the United States [1,2]. At present, the common treatments for CA include topical therapeutic agents, physical removal of warts, and photodynamic therapy (PDT) [3]. PDT is a non-invasive therapeutic method for many types of cancer as well as non-oncological diseases [4]. It uses a photosensitizer (PS) and a specific wavelength of light to produce reactive oxygen species (ROS), which results in cytotoxicity and cell death [5]. The cellular response to PDT is dependent on diverse factors, such as cell types, the concentration and localization of PS, and oxygen level. Previous studies have proved that the mechanisms of PDT-induced cell death include apoptosis, necrosis, and autophagy [6]. 5-aminolevulinic acid (ALA) is a second-generation PS that endogenously produces protoporphyrin IX (PpIX), thereby causing cell damage [7]. It has been demonstrated that ALA-PDT could promote apoptosis and autophagy to reduce HPV viral load [8]. Additionally, ALA-PDT could induce cell death via endoplasmic reticulum stress [9]. The imbalance between the production of ROS and the capability of cells to repair the damage can result in different mechanisms of cell demise. In addition to apoptosis, necrosis, and autophagy mentioned above, new forms of cell death such as necroptosis, ferroptosis, and pyroptosis have also been identified, typically in PDT-related research [10].

Pyroptosis is a new form of programmed cell death, triggered by inflammasomes, which play a major role in infectious diseases and immune defense [11]. Under the infectious condition, inflammasomes such as NOD-like receptor 3 (NLRP3) activate caspase-1, one of the inflammatory caspase family, which in turn cleave gasdermin D (GSDMD) [12,13]. The N-domain of GSDMD can specifically bind to phosphoinositides in the mammalian cell membrane and form membrane pores that trigger cell swelling and membrane rupture. Subsequently, the cells disperse intracellular contents and inflammatory cytokines interleukin (IL)-1β and IL-18, which contribute to inflammation response and innate immunity [13,14]. Some research has shown that the generation of cytotoxic ROS could induce cell pyroptosis and greater ROS production could promote PDT-induced pyroptosis ability [15]. Zhu et al. proved that sono-photodynamic therapy (SPDT) could trigger pyroptosis in HepG2 liver cancer cells [16] and Li et al. showed that PDT could induce pyroptosis in esophageal squamous cell carcinoma [17]. Furthermore, pyroptosis plays a significant role in the photodynamic ablation of cancer cells and may represent an alternative treatment for cancer [18]. However, the pyroptosis effect of ALA-PDT in treating HPV infection-related diseases warrants more research.

Therefore, we used HeLa cells (HPV-18 positive) as an in vitro model of HPV infection to investigate whether pyroptosis is the potential form of ALA-PDT killing HPV-infected cells and further explore the related mechanism. Our findings provide some clues indicating the mechanism of ALA-PDT in CA treatment.

## 2. Materials and Methods

### 2.1. Cell Culture

HeLa cells (HPV-18 positive) were obtained from the Clinical Research Center of Nanfang Hospital and cultured in high-glucose DMEM (Corning, NY, USA) supplemented with 10% fetal bovine serum (Excell, Shanghai, China) in a 37 °C incubator with 5% $CO_2$. ALA (Fudan Zhangjiang Bio-Pharm Co., Ltd., Shanghai, China) was dissolved in phosphate-buffered saline (PBS, Gibco, Waltham, MA, USA) and stored at −80 °C. Cells in logarithmic phase were transferred into target plates overnight and divided into the control, ALA, Light, and ALA-PDT groups. Then, the culture medium was replaced with serum-free medium for the control and Light group, and with serum-free medium supplemented with ALA in the dark for ALA and ALA-PDT groups. After 24 h, cells in the Light and ALA-PDT group were then restored with PBS and exposed to the laser at 635 nm wavelength. Subsequently, the liquids of four groups were replaced with fresh medium, followed by incubation for 24 h under dark conditions.

### 2.2. Preparation and Administration of MCC950

MCC950, a specific NLRP3 inhibitor, was purchased from MedChemExpress (Monmouth Junction, NJ, USA) and dissolved in serum-free DMEM. HeLa cells were pre-treated with 1 μm MCC950 and followed by laser irradiation.

### 2.3. Cell Viability Assay

Cell viability was determined by the cell counting kit-8 (CCK-8) assay (Fdbio science, Guangzhou, China). Cells were randomly seeded onto 96-well culture plates ($5 \times 10^3$ cells/well) overnight before treated with ALA-PDT. A total of 10 μL of CCK-8 solution was added to each well and the cells were incubated for 1 h in the dark. The absorbance of each well was measured at 450 nm by a microplate luminometer (Molecular Devices, San Jose, CA, USA).

### 2.4. Quantitative Real-Time PCR (qRT-PCR)

TRIzol reagent (TIANGEN, Beijing, China) was used to extract total RNA. After the detection of RNA concentration, RNA was reversely transcribed into cDNA using Color Reverse Transcription Kit (EZBioscience, Beijing, China). qRT-PCR was performed with $2 \times$ SYBR Green Color qPCR Mix (EZBioscience, Beijing, China). Primer sequences used were as follows: GAPDH forward, GGAGCGAGATCCCTCCAAAAT; GAPDH reverse,

GGCTGTTGTCATACTTCTCATGG; NLRP3 forward, GATCTTCGCTGCGATCAACAG; NLRP3 reverse, CGTGCATTATCTGAACCCCAC; Caspase-1 forward, TTTCCGCAAG-GTTCGATTTTCA; Caspase-1 reverse, GGCATCTGCGCTCTACCATC; GSDMD forward, GTGTGTCAACCTGTCTATCAAGG; GSDMD reverse, CATGGCATCGTAGAAGTGGAAG. The gene expression level was calculated by $2^{-\Delta\Delta CT}$ method, with GAPDH as the internal reference.

### 2.5. Western Blot

Cells were lysed with RIPA buffer (Leagene, Beijing, China), and total protein concentrations were measured by the bicinchoninic acid method using BCA Protein Assay Kit (Beyotime, Shanghai, China). The protein samples were separated by 12–15% sodium dodecyl sulfate–polyacrylamide gel for electrophoresis and then transferred to polyvinylidene fluoride membranes (Millipore, Burlington, MA, USA). The membrane was blocked with protein-free blocking solution for 20 min and then incubated at 4 °C overnight with the primary antibodies, including β-Tubulin (Abmart, Beijing, China, 1:2000), GSDMD (Proteintech, Wuhan, China, 1:500), caspase-1 (Proteintech, Wuhan, China, 1:1000), NLRP3 (Proteintech, Wuhan, China, 1:1000), IL-1β (Proteintech, Wuhan, China, 1:1000) and IL-18 (Proteintech, Wuhan, China, 1:1000). After three washes with 0.1% TBST, the membrane was incubated with horseradish peroxidase (HRP)-conjugated secondary antibodies (Bioss, 1:5000, Beijing, China) for 1.5 h at room temperature. Then the membrane was washed three times with 0.1% TBST. The protein bands were visualized using ELC hypersensitive luminescent liquid (Millipore, Burlington, MA, USA) and detected by a chemiluminescence apparatus (Tanon, Huzhou, China). Image J software was used to analyze the band intensity, using β-Tubulin as the internal control.

### 2.6. Intracellular Generation of ROS

The levels of Intracellular ROS were carried out with a ROS Assay Kit (Beyotime, Shanghai, China). After different treatments, the cells were incubated in DMEM with DCFH-DA for 20 min and washed with PBS. Images of cells were captured by an inverted light microscope (Olympus IX73, Tokyo, Japan).

### 2.7. Lactate Dehydrogenase (LDH) Release Assay

Pyroptosis was measured by the level of LDH released into the culture supernatants after different treatments. The release of LDH was detected using the LDH Cytotoxicity Assay kit (Beyotime, Shanghai, China) according to the manufacturer's protocol. The absorbance was measured at 490 nm and 630 nm with a microplate luminometer.

### 2.8. Hoechst 33342/PI Staining

To observe the morphological changes of cells, Hoechst 33342/PI Double Stain Kit (Solarbio, Bristol, UK) was used according to the manufacturer's instructions. The cells were resuspended in a binding buffer, followed by staining with Hoechst 33342 (5 µL/mL) and PI (5 µL/mL) for 30 min at 4 °C in the dark and washed by PBS. Microscopic images were obtained using an inverted light microscope.

### 2.9. Statistical Analysis

Data were presented as mean ± standard deviation, and each experiment was repeated at least in triplicate. GraphPad Prism software was used for statistical analysis. Data were analyzed using Student's *t*-test or one-way ANOVA. A value of $p < 0.05$ was considered statistically significant.

## 3. Results

### 3.1. Different Gradients of ALA and Energy Suppress HeLa Cell Proliferation

The effect of ALA-PDT on the viability of Hela cells is affected by the concentration of ALA and the energy of the laser. In order to figure out the optimal condition, the suppression was determined after diverse ALA-PDT treatments. We observed that the cell viability was energy-dependent in a certain range, from 0 to 4.8 J/cm$^2$ (Figure 1A). However, with a few exceptions, the maximum inhibition rate occurred when the concentration of ALA was 0.5 mM. The results showed that under the treatment with 0.5 mM ALA for 24 h and laser irradiation with the intensity of 1.2 J/cm$^2$, it was close to 50% inhibition (Figure 1B). Therefore, we carried out subsequent experiments according to these results.

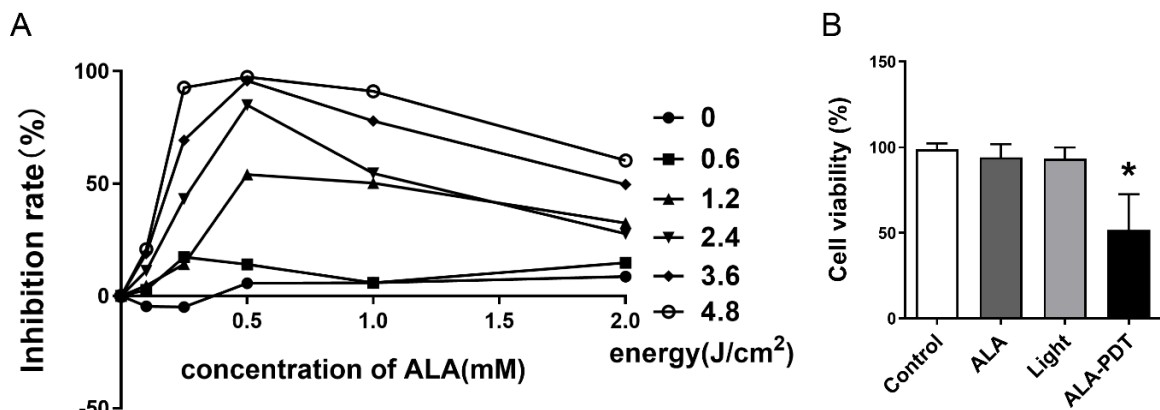

**Figure 1.** ALA-PDT inhibited the proliferation of HeLa cells. (**A**) Inhibitory effects of different gradients of ALA and laser irradiation energy on HeLa cells. (**B**) Inhibition rate of HeLa cells with 0.5 mM ALA and 1.2 J/cm$^2$ light energy. * $p < 0.05$ compared with the control group.

### 3.2. ALA-PDT Induces Pyroptosis in HeLa Cells

To investigate pyroptosis induced by ALA-PDT, the expression of pyroptosis-related key molecules was detected by qRT-PCR and western blot analysis. As shown in Figure 2A, the mRNA levels of NLRP3, caspase-1, and GSDMD in the ALA group and Light group showed no significant difference compared to that in the control group. The increasing levels of NLRP3, caspase-1, and GSDMD mRNA were documented in the ALA-PDT group. Western blot results showed that ALA-PDT elevated NLRP3, caspase-1, and GSDMD levels more than other groups, which was consistent with qPCR results. Moreover, the production of inflammatory factors IL-1β and IL-18 increased after being treated with ALA-PDT (Figure 2B–G).

### 3.3. ALA-PDT Enhances the ROS Production

Previous studies have demonstrated that ALA-PDT could induce oxidative stress to kill cells [19]. ROS is considered a critical factor mediating the activation of NLRP3 inflammasome and can trigger pyroptosis [20]. Therefore, we detected the intracellular ROS levels in HeLa cells with the treatment of ALA-PDT by evaluating the green fluorescence. Figure 3 showed that compared with the control group, the generation of ROS in the ALA-PDT group was significantly increased. In contrast, the green fluorescence of the ALA group and Light group were not significantly different. These results revealed that ALA-PDT could promote ROS production in HeLa cells.

### 3.4. ALA-PDT Induces Destruction of Cell Membrane of HeLa Cells

Pyroptosis can lead to damage to the cell membrane and LDH release was considered an important indicator of cell membrane integrity [21], so we carried out the LDH release assay. Figure 4A showed that the release of LDH in the ALA-PDT group significantly increased compared with other groups. To better understand the effect of ALA-PDT in

HeLa cells, we observed the morphologic change of cells under the microscopy and found that the majority of cells were swollen and lysed after ALA-PDT treatment (Figure 4B). Additionally, some cells had large bubbles blowing from the cell membrane. We also performed the Hoechst 33342/PI staining assay. As shown in Figure 4C,D, in contrast to the control group, the proportion of pyroptosis cell death (PI-positive cells) was significantly higher, which is consistent with the previous results.

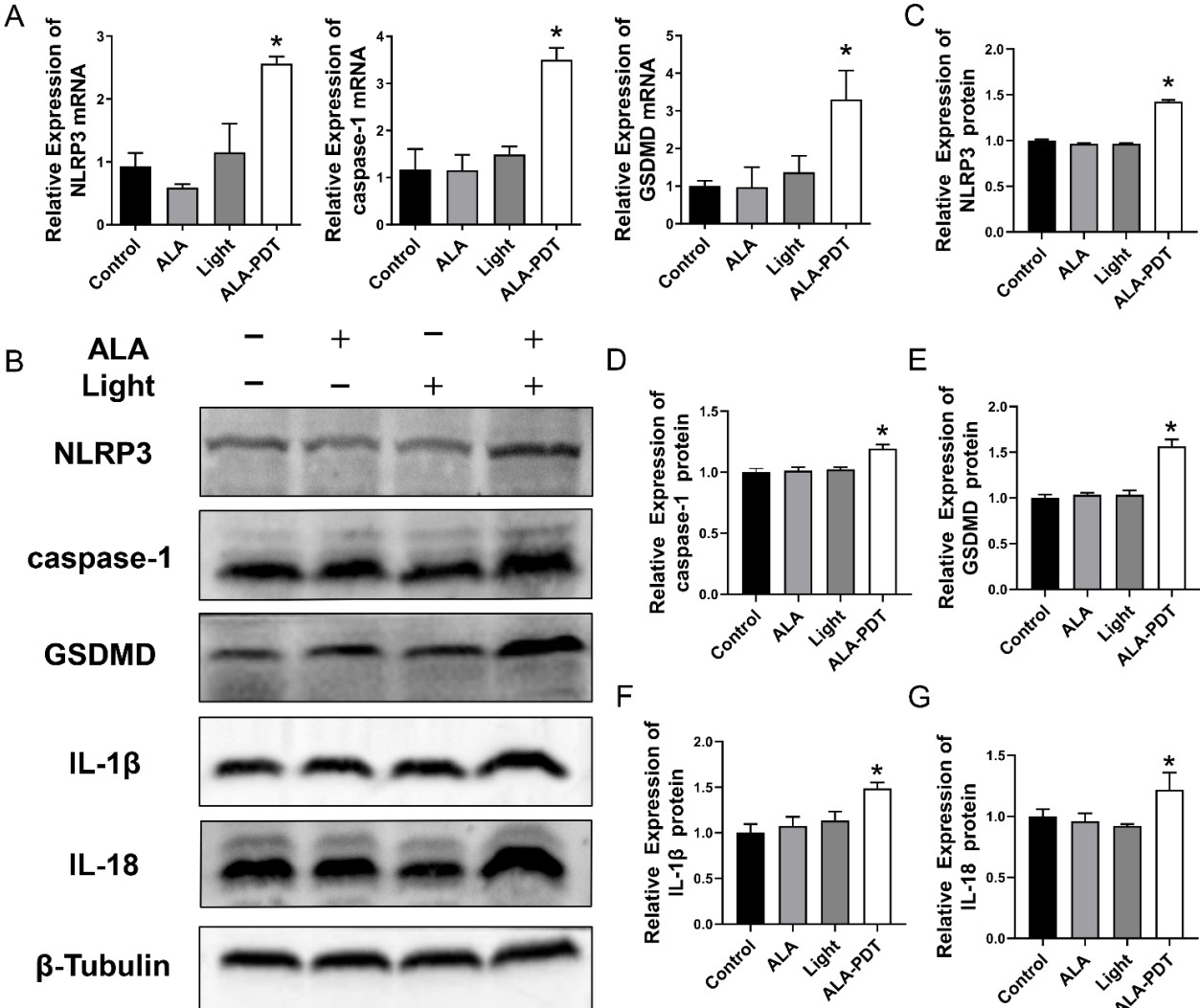

**Figure 2.** ALA-PDT induces pyroptosis in HeLa cells. (**A**) The mRNA expression of NLRP3, caspase-1, and GSDMD was detected by qPCR. * $p < 0.05$ compared with the control group. (**B**) The protein expression of NLRP3, caspase-1, GSDMD, IL-1β, and IL-18 was detected by western blot. (**C–G**) Statistical analysis of the protein levels of NLRP3, caspase-1, GSDMD, IL-1β, and IL-18. * $p < 0.05$ compared with the control group.

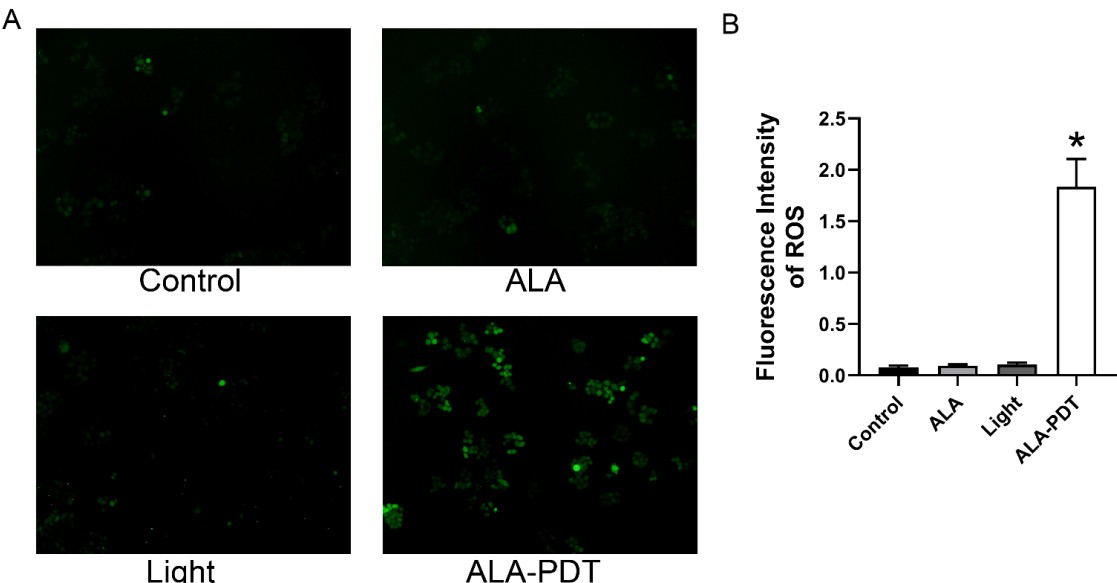

**Figure 3.** The effect of ALA-PDT on ROS production in HeLa cells. (**A**) The green fluorescence of ROS (100×). (**B**) Statistical analysis of the fluorescence intensity of ROS. * $p < 0.05$ compared with the control group.

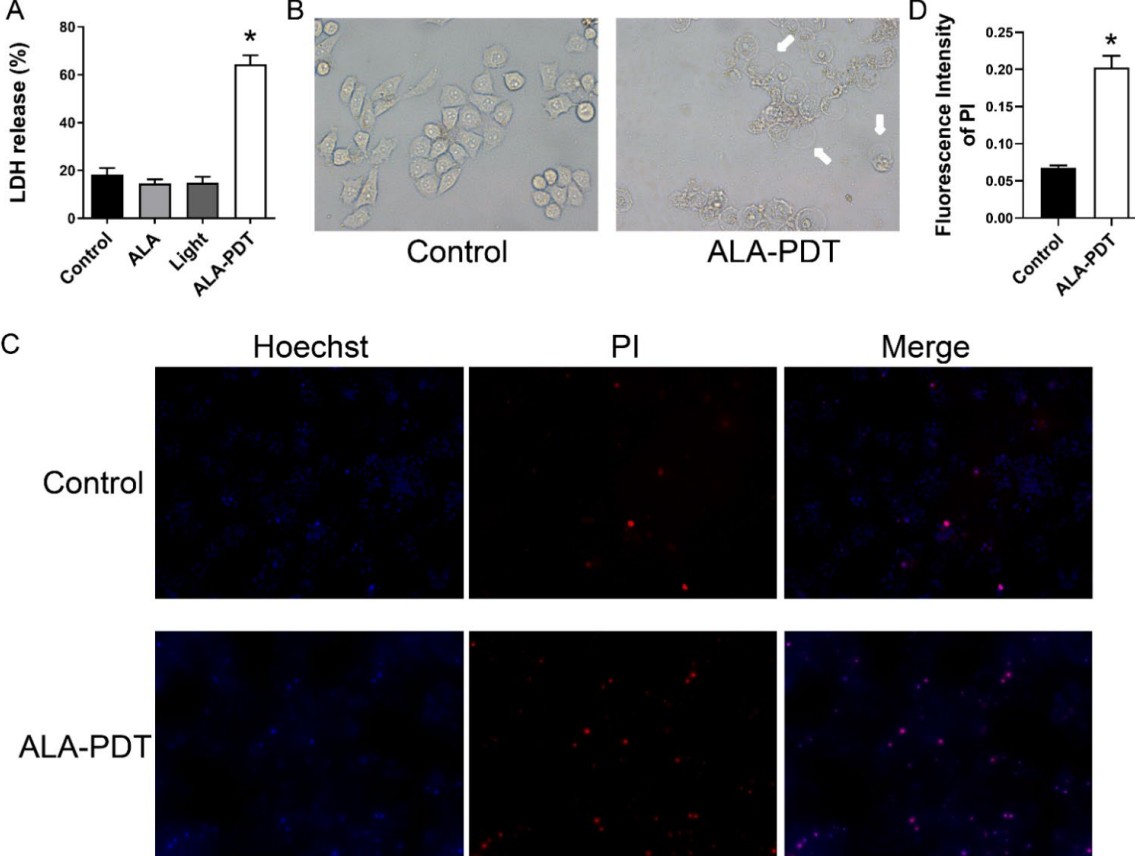

**Figure 4.** ALA-PDT induces the destruction of cell membrane and the morphologic change of HeLa cells. (**A**) ALA-PDT promoted LDH release in HeLa cells. * $p < 0.05$ compared with the control group. (**B**) The morphologic change of HeLa cells in bright field (400×). White arrowheads indicated the large bubbles emerging from the cell membrane. (**C**) Hoechst/ PI staining assay in HeLa cells after ALA-PDT treatment (100×). (**D**) Statistical analysis of the fluorescence intensity of PI-positive cells. * $p < 0.05$ compared with the control group.

### 3.5. MCC950 Alleviates the ALA-PDT-Induced Cell Death and ROS Production

To further investigate the effect of ALA-PDT-induced pyroptosis in Hela cells, cells were treated with MCC950, the NLRP3 specific inhibitor, and ALA-PDT to evaluate the cell viability. As shown in Figure 5A, MCC950 treatment improved the results of CCK8, suggesting that pyroptosis is potential cell death in ALA-PDT. We also evaluated the ROS level of cells and figured out that the use of MCC950 inhibited the generation of ROS (Figure 5B,C).

### 3.6. MCC950 Suppressed ALA-PDT-Induced Pyroptosis in HeLa Cells

In our qPCR analysis, decreased mRNA levels of NLRP3, caspase-1 and GSDMD were revealed in ALA-PDT combined MCC950 group (Figure 6A). Subsequently, we further examined the protein expression of NLRP3, caspase-1, GSDMD, and inflammatory mediators, IL-1β and IL-18 (Figure 6B–G). The results indicated that combined treatment decreased pyroptosis key molecule levels, indicating that MCC950 could alleviate the effect of ALA-PDT by mitigating pyroptosis-related death.

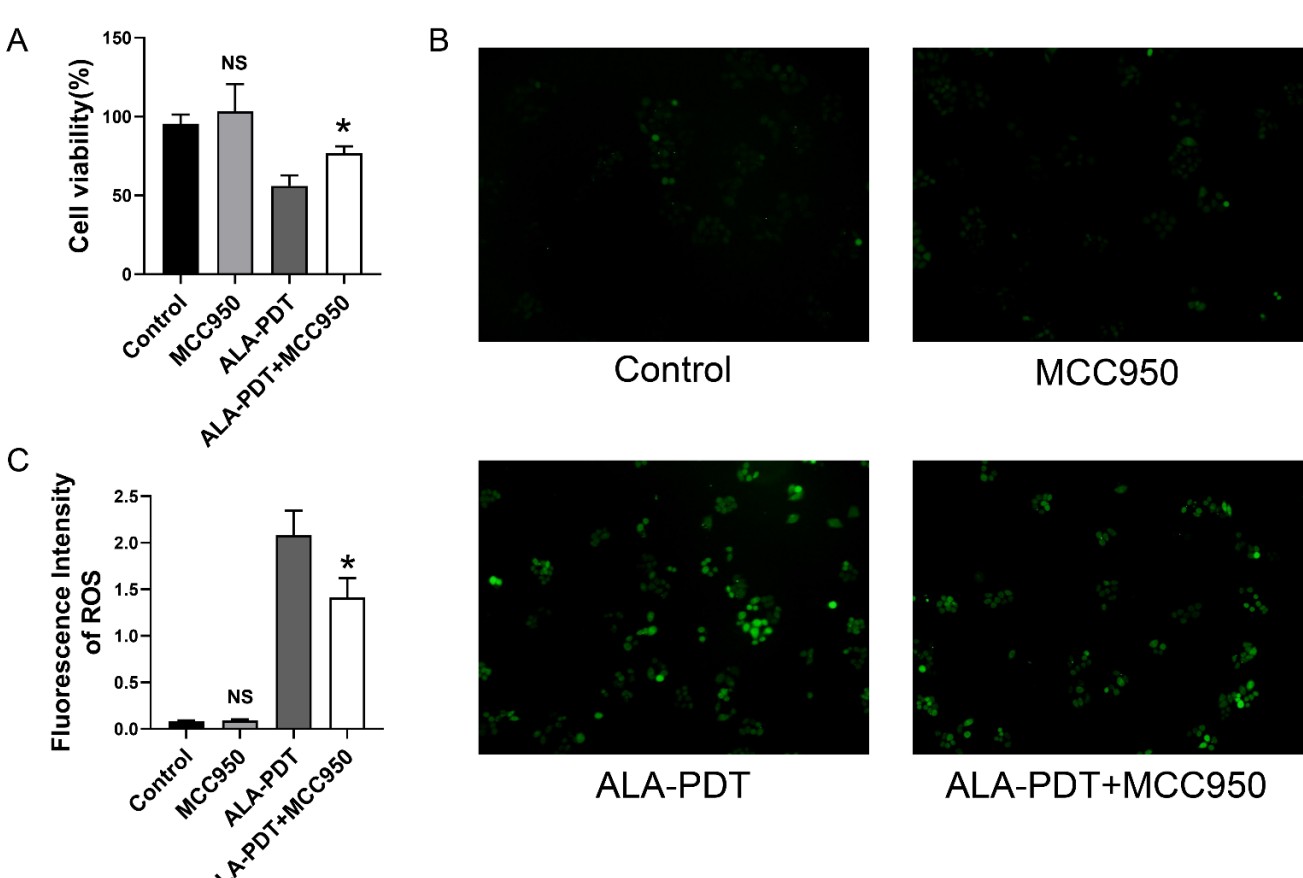

**Figure 5.** MCC950 suppressed ALA-PDT-induced cell death and ROS production. (**A**) CCK-8 assay for cell toxicity. NS: $p > 0.05$ compared with the control group. * $p < 0.05$ compared with ALA-PDT group. (**B**) The green fluorescence of ROS ($100\times$). (**C**) Statistical analysis of the fluorescence intensity of ROS. NS: $p > 0.05$ compared with the control group. * $p < 0.05$ compared with ALA-PDT group.

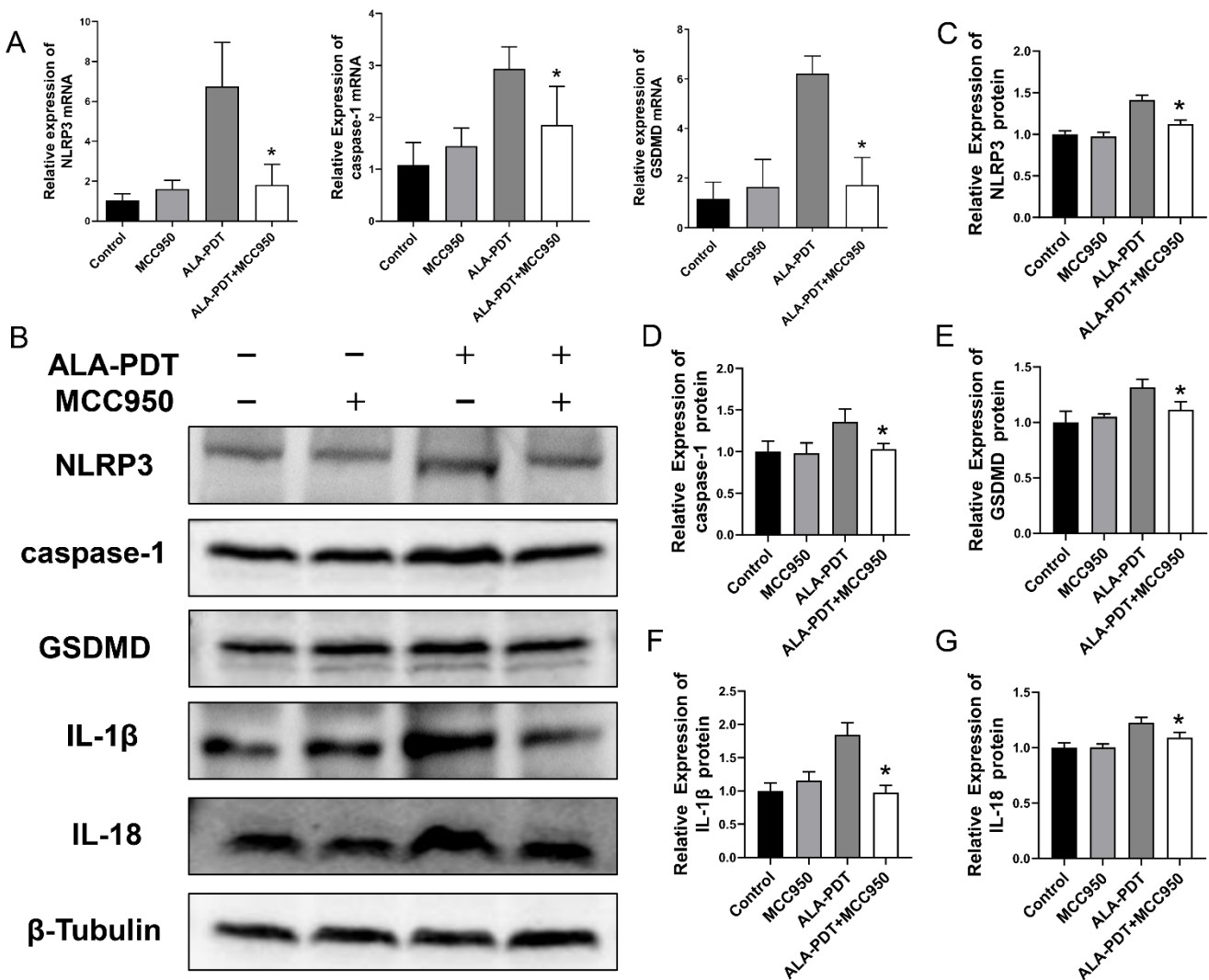

**Figure 6.** MCC950 suppressed ALA-PDT-induced pyroptosis. (**A**) The mRNA expression of NLRP3, caspase-1, and GSDMD was detected by qPCR. * $p < 0.05$ compared with ALA-PDT group. (**B**) The protein expression of NLRP3, caspase-1, GSDMD, IL-1β, and IL-18 was detected by western blot. (**C–G**) Statistical analysis of the protein levels of NLRP3, caspase-1, GSDMD, IL-1β, and IL-18. * $p < 0.05$ compared with ALA-PDT group.

### 3.7. MCC950 Abrogated the Pyroptotic Morphology in ALA-PDT-Treated Cells

We combined ALA-PDT with MCC950 and found that MCC950 treatment inhibited the liberation of LDH (Figure 7A). In addition, Figure 7B–D showed less membrane rupture and leakage after the treatment of MCC950. Results here showed that the use of pyroptosis inhibitor improved the pyroptotic microscopy and the formation of pyroptotic-like cells. However, both results of LDH release and PI staining showed that MCC950 cannot completely reverse the damage to the plasma membrane caused by ALA-PDT.

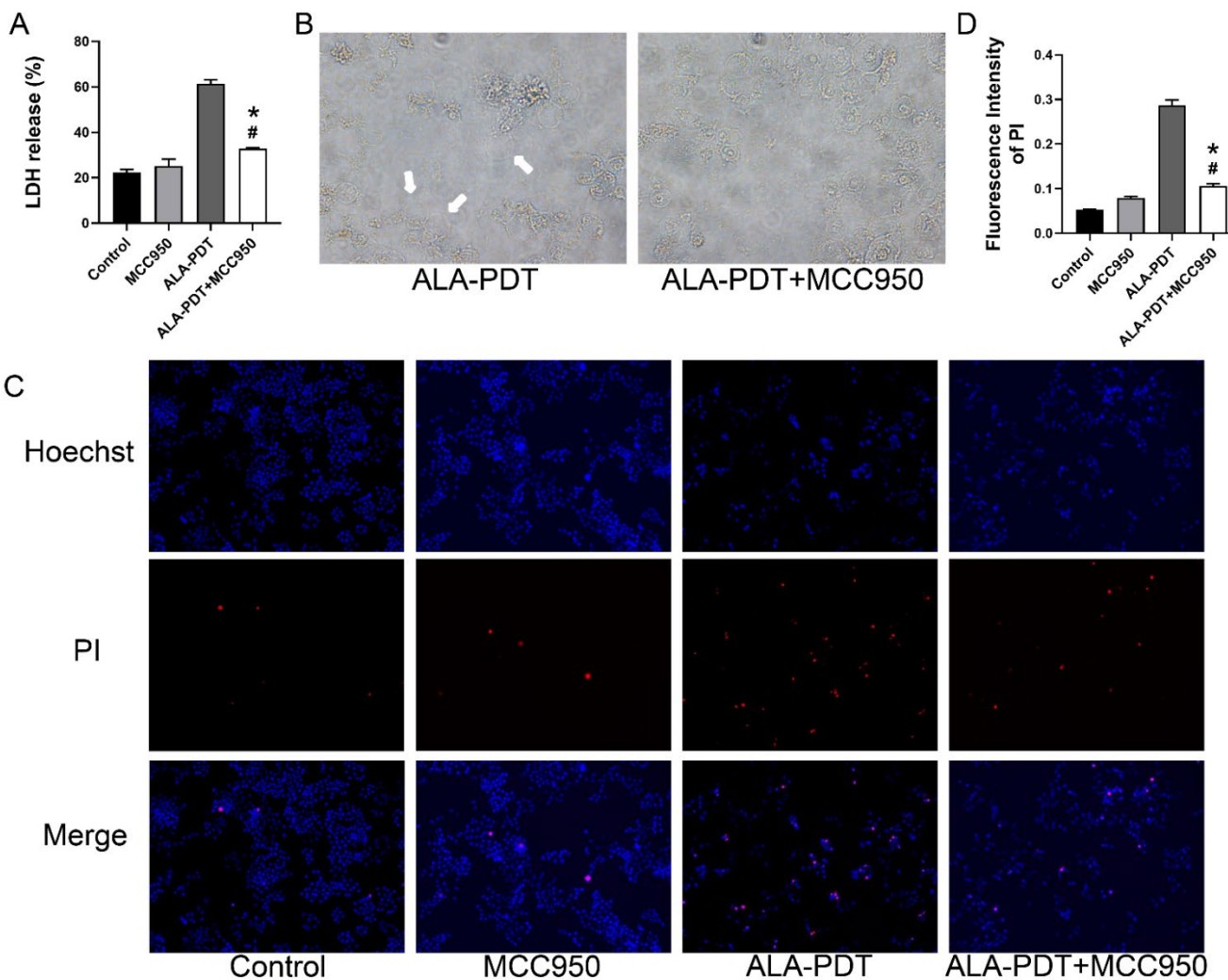

**Figure 7.** MCC950 abrogated the pyroptotic morphology induced by ALA-PDT in HeLa cells.
(**A**) ALA-PDT combined MCC950 reduced the release of LDH. * $p < 0.05$ compared with ALA-PDT
group. # $p < 0.05$ compared with the control group. (**B**) The morphologic change of HeLa cells after
ALA-PDT combined MCC950 in bright field (400×). White arrowheads indicated the large bubbles
emerging from the cell membrane. (**C**) Hoechst/ PI staining assay in HeLa cells after ALA-PDT
combined MCC950 (100×). (**D**) Statistical analysis of the fluorescence intensity of PI-positive cells.
* $p < 0.05$ compared with ALA-PDT group. # $p < 0.05$ compared with the control group.

## 4. Discussion and Conclusions

This study investigated the role of the inflammasome pathway in HPV-infected cells
induced by ALA-PDT. These findings showed the involvement of pyroptosis in ALA-PDT-
induced cell death and that MCC950 provided protection against this.

PDT has become a clinically approved treatment that has been proven to be effective
and safe to treat various skin and mucosal diseases [22]. Studies have reported that ALA-
PDT is an efficient and simple treatment modality to treat CA [23]. After being selectively
absorbed by HPV-infected cells, ALA transforms into PpIX and leads to the production of
ROS upon irradiation. Then, ROS causes cell damage and eventually destroys the entire
wart [23,24]. The modality for the response of cells to ALA-PDT has been extensively
studied [25,26]. However, the potential mechanism underlying ALA-PDT is still complex,
and the effect of ALA-PDT on pyroptosis warrants further research.

Pyroptosis is a new prospective manner to inhibit the viability and migration of tumor
cells [27]. Previous studies have identified that pyroptosis is a lytic form of inflammation-
related programmed cell death [28] and there are two main pathways of pyroptosis. The



classic pathway of pyroptosis is mediated by caspase-1 [29,30]. By contract, the non-canonical pyroptosis pathway is initiated by caspase-4 and caspase-5 in humans and caspase-11 in mice [31–33]. In addition to the methods mentioned above, early studies have shown that chemotherapy drugs or tumor necrosis factor (TNF) could cleave GSDME by activating caspase-3, a hallmark of apoptosis, which switches apoptosis to pyroptosis [34,35]. In our research, we demonstrated that ALA-PDT induced pyroptosis in HeLa cells, presenting some signs of pyroptosis, including increased the mRNA and protein levels of NLRP3, caspase-1, and GSDMD, and the secretion of inflammatory cytokines (IL-1β and IL-18). These results indicated that the canonical pathway of pyroptosis was activated after ALA-PDT.

The generation of large amounts of ROS could be the trigger for pyroptosis induced by ALA-PDT. In addition, it has been shown that the release of $Ca^{2+}$ is a key regulatory molecular of NLRP3 [36], and an increase in intracellular $Ca^{2+}$ concentration promotes the activation of NLRP3 [37]. Additionally, ALA-PDT could elevate the level of $Ca^{2+}$ in the cytoplasm [9], implying that there is more than one factor for ALA-PDT to trigger pyroptosis. NLRP3 is one of the best characterized inflammasomes that is involved in the pathogenesis and progression of many diseases [38]. Several NLRP3 inflammasome inhibitors have been developed and used in recent years, including MCC950, type I interferon (IFN), autophagy inducer, and microRNA [39]. MCC950 is a potent and selective inhibitor that blocks canonical and noncanonical activation of NLRP3 [40]. In our study, the role of pyroptosis in ALA-PDT was confirmed by the effect of MCC950. We found that MCC950 mitigated the incidence of pyroptosis and attenuated inflammation by inhibiting the expression of NLRP3. In addition, our data showed that MCC950 reduced the ROS generation induced by ALA-PDT, which may be related to the fact that MCC950 suppressed oxidative stress by inhibiting mitochondrial dysfunction [41,42].

In summary, our findings indicated that the activation of the canonical pathway of pyroptosis was involved in inflammatory and cell death induced by ALA-PDT; MCC950 was shown to facilitate these phenomena. Our study supported that ALA-PDT could promote cell death by regulating pyroptosis, which may present a new strategy to elevate the killing efficacy of ALA-PDT and provide support for the clinical treatment of HPV-related diseases.

**Author Contributions:** Writing—original draft preparation, J.W. and X.P.; writing—review and editing, S.W., M.X. and X.H.; software, H.L.; methodology, Y.Z., X.C. and Q.W.; supervision, K.Z. All authors have read and agreed to the published version of the manuscript.

**Funding:** This work was supported by the National Science Foundation of China (82073464).

**Institutional Review Board Statement:** Not applicable.

**Informed Consent Statement:** Not applicable.

**Data Availability Statement:** Not applicable.

**Conflicts of Interest:** The authors declare no conflict of interest.

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
