# Peer review of "The Effect of 5-Aminolevulinic Acid Photodynamic Therapy in Promoting Pyroptosis of HPV-Infected Cells"

_photonics, doi:10.3390/photonics9060408_

Round 1
Reviewer 1 Report
In Discussion part, the authors tried to explain the possible pyroptosis pathways in ALA-PDT which might involve caspase-1, caspase-3, caspase-4, caspase-5 and caspase-11. But only caspase-1 was tested in the study and then came to the conclusion that ALA-PDT was mediated in the classic pathway by caspase-1. We think this explanation was not conclusive enough.
In line 63, “warrant” is a verb here, please check the grammar.
In line 70, were these HeLa cells standard or HPV infected? If they were HPV positive, please describe them in this part.
In Figure 1A, please explain why ALA concentrations and laser irradiation were set from 0-2mM and 0-4.8J/cm2?
In Figure 2A, what is control group, ALA group and light group? These group classifications should be also mentioned in the Methods session above.
In Figure 2B, the variant above should be corrected from “PDT” to “Light”.
Reviewer 2 Report
The authors explore the mechanism of cell pyroptosis in HPV infected skin keratinocytes treated with ALA-PDT, which is innovative.
Main problems:
1. The authors used lactate dehydrogenase (LDH) experiment and Hoechst 33342 / PI staining experiment to verify the phenomenon of cell death induced by ALA-PDT. However, LDH and Hoechst 33342 / PI experiments can not accurately distinguish late apoptosis, necrosis and pyroptosis, which need to be further verified by more reasonable methods.
2. The authors found that the level of cellular ROS decreased after using NLRP3 inhibitor MCC950. After the use of inhibitors, whether they first inhibit cell pyroptosis and then reduce the level of ROS, or whether they reduce pyroptosis by clearing ROS, the author needs to explain the mechanism of action. In addition, does MCC950 have the effect of scavenging ROS?
3. It is suggested to use scanning electron microscope to observe the morphological changes of cells and pyroptosis.
Reviewer 3 Report
I recommended to accept this article in its present form
Reviewer 4 Report
The manuscript titled “ The effect of 5-aminolevulinic acid photodynamic therapy in 2 promoting pyroptosis of HPV-infected cells” is well written article, investigating the inflammation-based cell death mechanism of pyroptosis. The study carried out to demonstrate the role of the pyroptosis in PDT treatment using 5-ALA, holds potential in elucidating the complex mechanism behind ALA-PDT. However, there are some relevant controls that are missing in few experimental studies.
- For studies involving the NLRP3 inhibitor, MCC950, it will be important to include a “Dark” control with (ALA+MCC950) without light treatment to eliminate any additive effect playing a role.
- What was the rationale of choosing only HeLa cell lines for these studies? For studies involving investigation of cell death mechanisms, it is critical to perform such studies using multiple cell lines to validate and identify if the mechanisms are cell-dependent or not.
Round 2
Reviewer 2 Report
I think the authors responded to the reviewers' comments to a certain extent.
Although there is no further supplementary test, the current data can support the author's conclusion to a certain extent. I hope the author can improve the details concerned by the reviewers in the future research.